# Evaluation of Approaches for the Assessment of HER2 Expression in Breast Cancer by Radionuclide Imaging Using the Scaffold Protein [^99m^Tc]Tc-ADAPT6

**DOI:** 10.3390/pharmaceutics16040445

**Published:** 2024-03-23

**Authors:** Olga Bragina, Liubov Tashireva, Dmitriy Loos, Vladimir Chernov, Sophia Hober, Vladimir Tolmachev

**Affiliations:** 1Department of Nuclear Therapy and Diagnostics, Cancer Research Institute, Tomsk National Research Medical Center, Russian Academy of Sciences, Tomsk 634014, Russia; rungis@mail.ru (O.B.); chernov@tnimc.ru (V.C.); 2Research Centrum for Oncotheranostics, Research School of Chemistry and Applied Biomedical Sciences, Tomsk Polytechnic University, Tomsk 634009, Russia; tashireva@oncology.tomsk.ru; 3Laboratory of Molecular Cancer Therapy, Cancer Research Institute, Tomsk National Research Medical Center, Russian Academy of Sciences, Tomsk 634014, Russia; 4Department of General and Molecular Pathology, Tomsk National Research Medical Center, Tomsk 634014, Russia; loos.d@yandex.ru; 5Department of Protein Science, School of Engineering Sciences in Chemistry, Biotechnology and Health, KTH Royal Institute of Technology, 100 44 Stockholm, Sweden; sophia@kth.se; 6Department of Immunology, Genetics and Pathology, Uppsala University, 751 81 Uppsala, Sweden

**Keywords:** radionuclide molecular imaging, clinical study, HER2, scaffold protein, ADAPT6, technetium-99m

## Abstract

Due to its small size and high affinity binding, the engineered scaffold protein ADAPT6 is a promising targeting probe for radionuclide imaging of human epidermal growth factor receptor type 2 (HER2). In a Phase I clinical trial, [^99m^Tc]Tc-ADAPT6 demonstrated safety, tolerability and capacity to visualize HER2 expression in primary breast cancer. In this study, we aimed to select the optimal parameters for distinguishing between breast cancers with high and low expression of HER2 using [^99m^Tc]Tc-ADAPT6 in a planned Phase II study. HER2 expression was evaluated in primary tumours and metastatic axillary lymph nodes (mALNs). SPECT/CT imaging of twenty treatment-naive breast cancer patients was performed 2 h after injection of [^99m^Tc]Tc-ADAPT6. The imaging data were compared with the data concerning HER2 expression obtained by immunohistochemical evaluation of samples obtained by core biopsy. Maximum Standard Uptake Values (SUV_max_) afforded the best performance for both primary tumours and mALNs (areas under the receiver operating characteristic curve (ROC AUC) of 1.0 and 0.97, respectively). Lesion-to-spleen ratios provided somewhat lower performance. However, the ROC AUCs were still over 0.90 for both primary tumours and mALNs. Thus, lesion-to-spleen ratios should be further evaluated to find if these could be applied to imaging using stand-alone SPECT cameras that do not permit SUV calculations.

## 1. Introduction

Human epidermal growth factor receptor 2 (HER2) is overexpressed in substantial fractions of breast, gastric, ovarian, lung and bladder cancers [1]. Overexpression of HER2 and/or amplification of the *ERBB2* gene is observed in 15–20% of breast cancer patients and is associated with aggressive disease and a high risk of distant metastases [2]. It has been demonstrated that the use of the HER2-specific monoclonal antibody trastuzumab in combination with different chemotherapies increased survival when used for the treatment of metastatic breast cancer [3] or as adjuvant therapy [4]. Further progress was associated with the use of a combination of the HER2-specific monoclonal antibodies, trastuzumab and pertuzumab with docetaxel [5]. Also, the introduction of a conjugate of trastuzumab with a tubulin inhibitor emtansine increased the survival rate [6]. These treatments are all recommended for patients with HER2-positive breast cancer [7]. According to guidelines of the American Society of Clinical Oncology (2018), breast cancer is HER2-positive if an analysis of biopsy samples from tumours shows 3+ staining by immunohistochemistry (IHC) or *ERBB2* gene amplification of six or more copies via in situ hybridization (ISH) tests [8]. Recently, a breakthrough in HER2-targeted treatment was achieved with therapy utilizing trastuzumab deruxtecan (T-Dxd), which appeared to be effective not only in previously treated HER2-positive breast cancer [9] but also in breast cancer with lower HER2 expression [10]. These impressive results prompted a discussion if the levels of HER2 expression should be re-defined [10,11]. However, the current guidelines of the American Society of Clinical Oncology (ASCO) suggest the use of a combination of trastuzumab, pertuzumab and a taxane for first-line treatment and the use of T-Dxd if HER2-positive advanced breast cancer has progressed during or after first-line HER2-targeted therapy [7]. Furthermore, the expert panel (ASCO–College of American Pathologists) emphasized that “HER2 testing should still be optimized for the predictive purpose of identification of breast cancers with protein overexpression and/or gene amplification who could benefit from therapies aimed at disrupting HER2 signalling pathways” and stated that it is premature to change reporting terminology for lower levels of HER2 IHC expression [12]. Thus, the stratification of tumours into HER2-positive and HER2-negative, according to the existing classification, remains important for the selection of first-line treatment. Approximately 20–30% of breast cancer patients have regional or/and distant metastases at the time of diagnosis [13]. At the same time, there is a clinical problem of discordance in the HER2 status between the primary tumour and metastatic sites [14]. According to the results of a meta-analysis, which included 48 studies (1983–2012), pooled discordance proportions for HER2 were 8% (95% CI: 6–10%), while pooled proportions of tumours shifting from positive to negative and from negative to positive were 13% and 5% (*p* = 0.0004), respectively [15]. Sanchar and co-workers demonstrated that additional biopsies of metastatic sites were associated with improved survival (HR = 0.67. *p* = 0.002) [16].

Despite the importance of morphological and immunohistochemical metastatic verification, repetitive sampling is often difficult to implement due to anatomical positions of metastases, possible complications after core biopsies or the patient’s decline. Radionuclide molecular imaging of HER2 expression is an emerging non-invasive approach for the stratification of patients for targeted therapy due to the possibility of HER2 mapping in the case of locally advanced and metastatic breast cancer. Further, the method offers a possibility to perform repeated follow-up studies during the cancer treatment [17].

Currently, several formats of targeting probes for the imaging of HER2 have been evaluated in preclinical research and in clinics [17,18]. A very promising class of such probes are non-immunoglobulin engineered scaffold proteins (ESP). ESPs have a low molecular weight (between 4 and 19 kDa) and high affinities (below 5 nM). Further, unlike full-length monoclonal antibodies, ESP-based agents enable high-contrast clinical imaging within 2–4 h after injection [17,18]. 

Albumin-binding domain-derived affinity proteins (ADAPTs) are engineered scaffold proteins, which are small (5–7 kDa), and display high affinity and specificity to selected targets [19]. In preclinical studies, an ADAPT labelled with ^99m^Tc ([^99m^Tc]Tc-ADAPT6) demonstrated efficient differentiation of HER2-positive and HER2-negative tumours, and low uptake in tissues, which frequently harbour breast cancer metastases [20,21].

A Phase I clinical evaluation of [^99m^Tc]Tc-ADAPT6 in primary breast cancer patients (ClinicalTrials.gov identifier NCT03991260) showed the safety of this tracer. The effective dose of 0.009 ± 0.002 mSv/MBq suggested that multiple imaging procedures using [^99m^Tc]Tc-ADAPT6 would be permissible [22]. The use of tumour-to-contralateral site ratios 2 h after injection permitted discrimination between HER2-positive and HER2-negative primary breast tumours (*p* < 0.001, Mann–Whitney test) when the optimal ADAPT6 mass (500 µg) was injected [22]. Furthermore, a direct comparison demonstrated that [^99m^Tc]Tc-ADAPT6 provides higher uptake (SUV_max_) in HER2-positive breast cancer lesions than ^99m^Tc-labeled DARPin G3 [23], when both tracers were injected with an optimal mass and imaging was performed at the optimal time point for each tracer. The promising results of these studies prompt further clinical development. Hence, [^99m^Tc]Tc-ADAPT6 was selected for further clinical evaluation.

The Phase I trial focused on primary tumours because biopsy samples are routinely taken from these tumours and their HER2 status is evaluated to determine eligibility for first-line HER2-targeting therapy. To increase the efficacy of the treatment, it would be desirable to understand if [^99m^Tc]Tc-ADAPT6-based imaging enables the determination of HER2 status in metastatic lesions. A possible model for such lesions would be axillary lymph node metastases because taking biopsy samples for imaging data verification is less invasive compared to, e.g., bone, lung or liver metastases.

The Phase I study was performed using a stand-alone SPECT scanner without CT. Thus, the use of the tumour-to-reference ratio was the only option for the assessment of [^99m^Tc]Tc-ADAPT6 uptake in HER2-positive and HER2-negative tumours. Modern scanners permit co-registration of SPECT and CT data and the use of CT data enables accurate correction of SPECT scans for attenuation and scattering and thereby ensures the accuracy of activity concentration measurement in vivo, which is similar to the accuracy of PET [24]. Application of semi-quantitative analysis using Standard Uptake Values (SUVs) has been extensively investigated in SPECT/CT-based molecular imaging, see, e.g., [25,26,27,28,29]. Therefore, the implementation of such an analysis in the planned Phase II trial would be appropriate. However, it would be desirable to evaluate the feasibility of such an approach before including it in the protocol of a large multicentre trial.

The primary objectives of this study (ClinicalTrials.gov Identifier: NCT05412446) were the evaluation of uptake of [^99m^Tc]Tc-ADAPT6 (SUV) and lesion-to-background ratios for primary tumours and metastatic lymph nodes in patients with HER2-positive and HER2-negative breast cancer. 

The secondary objective was to compare the imaging data with the data concerning HER2 expression obtained by immunohistochemistry (IHC) and/or fluorescent in situ hybridization (FISH) analysis of biopsy samples in therapy-naïve patients.

Maximum Standard Uptake Values (SUV_max_) were selected for the analysis because phantom experiments demonstrated that this metric provides the most accurate quantification for SPECT/CT measurements [24]. Contralateral sites, spleen and latissimus dorsi muscle were evaluated as references since previous studies suggested that their use as reference tissue provided the best results for the imaging of HER2 expression using ^111^In- and ^68^Ga-labelled Affibody molecules [30].

## 2. Materials and Methods

This was a prospective, open-label, non-randomized, single-centre study. The Scientific Council of Cancer Research Institute and Board of Medical Ethics and Tomsk National Research Medical Centre of the Russian Academy of Sciences approved the protocol of the study (No. 4. 4 March 2022). Informed consent forms were signed in all cases before inclusion in the study.

Twenty, untreated (systemic therapy or local treatment) breast cancer patients (T_2–4_N_1–3_M_0–1_) with metastatic axillary lymph nodes (mALN) were included in the study (Figure 1). Eligible patients were adults (from 36 to 67 years) and had a performance status score of 0 or 1 on the Eastern Cooperative Oncology group scale (ECOG). According to HER2 expression in primary tumours and mALNs, patients were divided into two groups: 12 breast cancer patients with HER2 overexpression of primary tumours and 8 patients with negative HER2 expression (Table 1).

Patients were excluded if they had been treated with any systemic therapy (chemo-/targeted therapy) before the study or demonstrated a second, non-breast malignancy, active current autoimmune disease, history of autoimmune disease, active infection, history of severe infection within the previous 3 months (if clinically relevant at screening), a known positive HIV test, chronically active hepatitis B or C, the administration of other investigational medicinal products within 30 days of screening or ongoing toxicity > grade 2 from previous standard or investigational therapies according to the US National Cancer Institute (eligibility criteria are provided in Appendix A).

All patients underwent additional examinations before being included in the study. Mammography (Giotto Image, Sasso Marconi, Italy) and ultrasound of breast and regional lymph nodes (GE LOGIQ E9, Chicago, IL, USA) were performed as a local standard for assessing regional tumour spread. Bone scans (Siemens Symbia Intevo Bold, Siemens Healthineers, Erlangen, Germany) with ^99m^Tc-pyrophosphate, chest computed tomography (CT) (Siemens Somatom Emotions 16 ECO, Siemens Healthineers, Erlangen, Germany) and ultrasound of the liver (GE LOGIQ E9) were used to evaluate conditions of distant organs. The size of the primary breast tumours and metastatic lymph nodes was measured by ultrasound. CT of abdominal organs and brain magnetic resonance imaging were conducted in cases where suspected metastatic processes occurred.

### 2.1. Immunohistochemistry Analysis

In all patients, the core biopsies of the primary tumours and the mALNs were performed under ultrasound guidance and the HER2 expression was evaluated by immunohistochemistry (IHC). Formalin-fixed paraffin-embedded sections (7 µm) were stained by VENTANA anti-HER-2/neu (4B5) Rabbit Monoclonal Primary Antibodies using the Ventana Benchmark Ultra Instrument (Roche, Basel, Switzerland) according to the manufacturer’s instructions. HER2 expression was scored according to the guidelines of the American Society of Clinical Oncology and the College of American Pathologists (ASCO/CAP2018) [8]. A score of 3+ by IHC was defined as HER2-positive status. In cases of equivocal IHC status, the FISH test (HER2/CEP17 FISH probes, Kreathech, Amsterdam, The Netherlands) was performed according to the manufacturer’s instructions. HER2 status was considered negative in cases of a score of 0 and 1+ by IHC or score 2+ and FISH-negative.

### 2.2. Purification and Labelling of ADAPT6

ADAPT6 protein was produced in E. coli and purified using immobilized metal ion chromatography as described in [20]. Analysis by liquid chromatography–electrospray ionization mass spectrometry (6520 Accurate Q-TOF LC/MS, Agilent, Santa Clara, CA, USA) confirmed the identity of the protein (measured Mw 6954 Da, calculated Mw 6954.7 Da). No impurities were detected by reversed-phase HPLC (RP-HPLC) using the Zorbax 300SB-C18 column (4.6 × 150 mm, 3.5 μm particle size, Agilent), i.e., chemical purity was 100%. The levels of endotoxins (0.49 EU/mg freeze-dried protein) and residual *E coli* proteins (31.8 ng/mg of freeze-dried protein) were very low and met the requirements of European Pharmacopeia. Aliquots containing 500 µg ADAPT6 were prepared and freeze-dried.

Test labelling of qualification batches was performed according to the protocol described below. The identity of ADAPT6 labelled with ^99m^Tc was confirmed by radio-RP-HPLC (Phenomenex LC Luna 5 µm C18 column, 150 × 4.6 mm, 100 Å particle size, Danaher, Washington, DC, USA). Specific binding of the clinical batch of [^99m^Tc]Tc-ADAPT6 to HER2-expressing cancer cells was confirmed by in vitro saturation assay, as described in [20]. Sterility and endotoxin levels were evaluated according to the European Pharmacopoeia after decay of ^99m^Tc. According to national guidelines for conducting preclinical studies of drugs, the single-dose toxicity after intravenous injection was determined in mice and rats. No toxic effects were observed. 

Radiolabelling was performed in a GMP-compliant way at the Department of Radionuclide Therapy and Diagnostics, Tomsk Cancer Research Institute, according to national regulations. Freeze-dried ADAPT6 (500 µg) was reconstituted by adding sterile sodium phosphate buffer, at pH 7.5 (100 µL), using a sterile syringe followed by incubation for 30 min at room temperature. An eluate from a generator of ^99m^Tc (500 µL) was added to a sealed vial containing the CRS kit (Centrum for Radiopharmaceutical Sciences, Willigen, Switzerland) and incubated at 100 °C for 30 min. After incubation, 400 μL of the resulting solution was transferred by a sterile syringe to the vial containing reconstituted ADAPT6, followed by incubation for 60 min at 50 °C. [^99m^Tc]Tc-ADAPT6 was purified by size exclusion chromatography using sterilized NAP-5 columns (Sephadex G-25, GE, Healthcare, Chicago, IL, USA) pre-equilibrated and eluted with sterile sodium phosphate buffer. The purified fraction was brought to a volume of 10 mL using a sterile isotonic NaCl solution. 

A small aliquot was taken for analysis of pH and radiochemical purity. The pH of the drug product was determined using pH test strips. Routine analysis of the radiochemical purity was performed using instant thin layer chromatography (Agilent Technologies, Santa Clara, CA, USA). The mobile phases were PBS (Rf = 0 for [^99m^Tc]Tc-ADAPT6 and [^99m^Tc]TcO_2_; Rf = 1 for [^99m^Tc]Tc(H_2_O)_3_(CO)^3+^ and [^99m^Tc]TcO) and pyridine:acetic acid:water at 10:6:3 (Rf = 0 for [^99m^Tc]TcO_2_ and Rf = 1 for the [^99m^Tc]Tc-ADAPT6, [^99m^Tc]Tc(H_2_O)_3_(CO)^3^ and [^99m^Tc]TcO^4−^). The bubble-point method was used to test the filter integrity. 

A visual inspection was performed. The solution was clear, non-opalescent and colourless. The radiochemical purity of [^99m^Tc]Tc-ADAPT6 was 97.6 ± 1.4%. The pH was 7.4. The acceptance criteria of radiochemical purity, activity concentration, activity, pH, colour/transparency, and endotoxin level were met. 

### 2.3. Imaging Protocol

Imaging was performed 2 h after injection of [^99m^Tc]Tc-ADAPT6 using a Siemens Symbia Intevo Bold scanner with a high-resolution low-energy collimator. In all breast cancer patients, SPECT/CT scans (SPECT: 60 projections of 20 s each; images stored in a 256 × 256 pixel matrix; CT: 130 kV; 36 mAs) of the chest were completed and reconstructed using the reconstruction xSPECT (Siemens) protocol based on the ordered subset conjugate gradient (OSCG) method (24 iterations with 2 subsets). The 3D Gaussian FWHM 10 mm filter (Soft Tissue) was used. The images were processed using the proprietary software (Version 2006A) package Syngo.via Siemens Healthineers, Erlangen, Germany).

Maximum Standard Uptake Values were normalized to the participants’ body weight (SUV_max_) and calculated for primary tumours, contralateral symmetric breast regions and mALN and contralateral symmetric lymph node (LN) regions 2 h after injection. Additionally, SUV_max_ was also calculated for the liver, latissimus dorci muscle (LDM) and spleen for determination of the best tumour-to-reference and mALN-to-reference tissue ratio.

### 2.4. Statistics

Values are reported as mean ± SD. Statistical analysis was performed using Prism 10.2 for Windows (GraphPad Software, LLC, San Diego, CA, USA). The nonparametric Mann–Whitney *U* test was used to determine whether the differences between values for HER2-positive and HER2-negative tumours were significant. A 2-sided *p* value of less than 0.05 was considered significant.

## 3. Results

### 3.1. Biopsies and HER2 Status in LN Metastases

All 20 biopsies from 20 patients confirmed breast cancer metastases in ALN. According to the IHC results, there was no difference in HER2 status between primary breast tumours and mALNs (Table 1).

### 3.2. Compliance with Labelling and Imaging Protocols

ADAPT6 protein was labelled with ^99m^Tc according to cGMP immediately before the intravenous injection. The radiochemical purity of [^99m^Tc]Tc-ADAPT6 was 97.6 ± 1.4%. The average injected activity of [^99m^Tc]Tc-ADAPT6 was 447 ± 225 MBq. The SPECT/CT scans were performed according to the protocol for all twenty breast cancer patients.

### 3.3. Uptake of [^99m^Tc]Tc-ADAPT6 in Primary Tumour and Tumour-to-Contralateral Ratios

All tumours and mALNs were visualized using [^99m^Tc]Tc-ADAPT6 2h after injection in both HER2-positive and HER2-negative cases (Figure 2 and Figure 3). 

The results of this study show that SUV_max_ (Appendix A) was significantly higher in HER2-positive primary breast tumours (8.8 ± 2.7) compared to HER2-negative tumours (3.5 ± 1.4) (*p* < 0.0001, Mann–Whitney test). The difference in the tumour-to-contralateral ratio between HER2-positive (17.7 ± 9.54) and HER2-negative tumours (6.6 ± 2.7) was also significant (*p* = 0.0007, Mann–Whitney test) (Figure 4).

### 3.4. Uptake of [^99m^Tc]Tc-ADAPT6 in Reference Organs and Tumour-to-Reference Tissue Ratios

The uptake (SUV_max)_ of [^99m^Tc]Tc-ADAPT6 in the liver, LDM and spleen 2 h after injection of [^99m^Tc]Tc-ADAPT6 were 3.2 ± 1.2, 0.42 ± 0.15, 1.4 ± 0.7, respectively, in the case of HER2-positive tumours and 2.8 ± 0.6, 0.43 ± 0.2 and 1.5 ± 0.3 in the case of HER2-negative breast tumours. The difference between [^99m^Tc]Tc-ADAPT6 uptake in each reference organ in patients with HER2-positive and HER2-negative tumours was not significant (*p* > 0.05, Mann–Whitney test). Tumour*-*to*-*liver, tumour*-*to*-*LDM and tumour*-*to*-*spleen ratios were considerably higher in the case of HER2-positive tumours (2.8 ± 0.51, 26.1 ± 20.1 and 7.3 ± 4.02, respectively) than in HER2-negative ones (1.2 ± 0.5, 9.1 ± 3.9 and 2.4 ± 1.2, respectively) (*p* < 0.005, Mann–Whitney test) (Appendix A and Appendix A).

### 3.5. [^99m^Tc] Tc-ADAPT6 in Metastatic Axillary Lymph Nodes, mALN-to-Contralateral and mALN-to-Reference Tissue Ratios

SUVs_max_ were considerably higher in mALNs with HER2 overexpression (8.7 ± 4.6) compared to HER2-negative mALNs (2.6 ± 0.9) (*p* = 0.0001, Mann–Whitney test). In a similar manner to the data for the primary tumours, mALN*-*to*-*contralateral ratios were higher in HER2-positive mALNs (34.3 ± 27.6) than in HER2-negative mALNs (9.2 ± 3.5) (*p* = 0.0005, Mann–Whitney test) (Figure 5, Appendix A).

Tumour-to-liver, tumour*-*to*-*LDM and tumour*-*to*-*spleen ratios in mALNs with HER2 overexpression were 2.8 ± 1.3, 27.3 ± 28.0, and 6.7 ± 2.9, respectively. Tumour-to-liver, tumour*-*to*-*LDM and tumour*-*to*-*spleen ratios in HER2-negative nodes were 0.9 ± 0.4. 6.7 ± 2.9 and 1.8 ± 0.9, respectively. All three parameters were much higher in patients with HER2-positive patients (*p* < 0.003, Mann–Whitney test) (Appendix A).

### 3.6. Determination of the Most Valuable Indicator for the Detection of HER2 Overexpression in Primary Breast Tumours Using [^99m^Tc] Tc-ADAPT6

The use of SUV_max_ of primary tumour*,* tumour-to-contralateral, tumour-to-liver, tumour*-*to*-*LDM and tumour-to-spleen ratios for the discrimination of HER2-positive and HER2-negative lesions was evaluated using ROC analysis. In model building, the SUV_max_ of the primary tumour demonstrated the best performance with an AUC of 1.00 (95% CI of 1.00 to 1.00), sensitivity of 100% and specificity of 100% with a cut off value > 5.36 (Figure 6A). Corresponding values for the other parameters were as follows: tumour*-*to*-*contralateral ratio AUC of 0.92 (95% CI of 0.80 to 1.00), sensitivity of 91.67% and specificity of 87.50%; a cut off value > 8.61, tumour*-*to*-*liver ratio AUC of 0.98 (95% CI of 0.92 to 1.00), sensitivity of 91.67% and specificity of 100%; a cut off value > 2.50, tumour*-*to*-*LDM ratio AUC of 0.94 (95% CI of 0.85 to 1.00), sensitivity of 83.33% and specificity of 100%; cut off > 14.94 and tumour*-*to*-*spleen ratio AUC of 0.96 (95% CI of 0.87 to 1.00), sensitivity of 100.00% and specificity of 87.50% with a cut off value > 3.59 (Appendix A).

### 3.7. Determination of the Most Valuable Indicator for the Detection of HER2 Overexpression in Metastatic Axillary Lymph Nodes Using [^99m^Tc] Tc-ADAPT6 in Breast Cancer Patients

Analysis of the parameters associated with metastatic axillary lymph nodes demonstrated the best discrimination of HER2-positive and HER2-negative lesions when using the SUV_max_ of mALNs with an AUC of 0.97 (95% CI of 0.89 to 1.00), sensitivity of 91.67% and specificity of 100% with a cut off value > 4.22 (Figure 6) or an mALN*-*to*-*contralateral ratio AUC of 0.93 (95% CI of 0.83 to 1.00), sensitivity of 91.67% and specificity of 87.50% with a cut off > 13.00. The use of ratios between mALNs and other reference tissues provided lower sensitivity and specificity, demonstrated by the following results: an mALN*-*to*-*liver ratio AUC of 0.94 (95% CI of 0.85 to 1.00), sensitivity of 83.33% and specificity of 87.50% with a cut off > 1.465, mALN*-*to*-*LDM ratio AUC of 0.89 (95% CI of 0.75 to 1.00), sensitivity of 83.33% and specificity of 87.50% with a cut off > 10.37 and mALN*-*to*-*spleen ratio AUC of 0.94 (95% CI of 0.85 to 1.00), sensitivity of 83.33% and specificity of 100% with a cut off > 4.724 (Appendix A).

## 4. Discussion

Scaffold proteins are a broad group based on structurally different frameworks [31]. The major common feature of different scaffold proteins is a small molecular weight, which is favourable for tumour targeting in vivo [32]. The ESPs differ in structure, size and amino acids exposed on their surfaces and the affinities of these proteins to the same target might differ quite substantially, i.e., up to two orders of magnitude. Accordingly, the uptake of different scaffold proteins in tumours with the same target expression level, as well as uptake in normal organs, might vary appreciably, which affects the accuracy of the diagnostics. At this stage, an accumulation of clinical data is important to select the best diagnostic probe for each application. Since an initial evaluation suggested that [^99m^Tc]Tc-ADAPT6 provides a higher uptake (SUV_max_) in HER2-positive breast cancer lesions than ^99m^Tc-labeled DARPin G3 [23], [^99m^Tc]Tc-ADAPT6 was selected for further clinical evaluation. 

The optimal Phase II study protocol is critical for obtaining essential information concerning the sensitivity and specificity of [^99m^Tc]Tc-ADAPT6 and the selection of criteria for the discrimination of HER2-positive and HER2-negative lesions would affect the study outcome. The use of SUVs offers a standardized and semi-quantitative assessment of radiotracer tumour uptake, making comparisons and interpretations more consistent between different observers. Also, the accuracy of SUV measurements in SPECT/CT imaging can be expected to improve data analysis, achieving further development of technology and methodology. However, algorithms for attenuation and scatter and recovery correction methods might be different between cameras from different manufacturers, which are installed in participating clinical centres. A lack of standards for quantitative SPECT imaging might make it challenging to establish uniform protocols and threshold values for SUV. Furthermore, SUV measurements might be affected by the accuracy of the measurement or recording of injected activity and patient weight, and by the calibration of the camera.

Tumour-to-reference ratios are widely applied in PET and SPECT imaging and are a metric for quantifying the relative uptake of a radiotracer in a suspected lesion compared to the background or reference normal tissue. In the case of using an intra-image reference, this method is robust, compensating for suboptimal camera calibration or activity measurements. This method was successfully applied in earlier Phase I evaluations of [^99m^Tc]Tc-ADAPT6 and [^99m^Tc]Tc-DARPin G3 [22,33]. It has been shown that the use of tumour-to-spleen ratios permitted an accurate discrimination between HER2-positive and HER2-negative tumours using ^111^In- and ^68^Ga-labelled affibody molecules [30]. Importantly, the accuracy of this approach depends on the appropriate selection of the reference tissue. Sandberg and co-authors [30] have formulated the following criteria for the selection of a reference tissue for the determination of HER2 status by affibody-mediated imaging: (1) correlation with data from biopsy analysis; (2) low variation in tracer uptake; (3) low probability of homing breast cancer metastases. Using these criteria, Sandberg and co-authors [30] proposed to use the spleen as a reference tissue, indicating spine muscle as the second-best choice. These tissues, along with a contralateral site and the liver, were also evaluated in this study.

Another potential issue in imaging is the difference in the uptake in the primary tumours and metastases due to, e.g., differences in the perfusion level. Zhou and co-authors demonstrated a substantial difference between the uptake of the ^68^Ga-labelled affibody in HER2-positive primary gastric cancer and its metastases in different organs, which were dependent on the homing organ [34]. The authors suggested the introduction of different SUV cut-off values for the discrimination of HER2-positive and negative metastases in different organs. Similarly, Bragina and co-authors [26] observed a noticeably higher uptake of ^99m^Tc-labeled affibodies in hepatic metastases compared to primary breast tumours. Hence, we took into consideration that this might also be the case for [^99m^Tc]Tc-ADAPT6. Therefore, patients with lymph node metastases were included in this study.

The results of this study have demonstrated that the uptake (SUV_max_) of [^99m^Tc]Tc-ADAPT6 in HER2-positive primary tumours (8.8 ± 2.7) did not differ significantly (*p* > 0.05, paired *t*-test) from its uptake in the corresponding lymph node metastases (8.7 ± 4.6). In both cases, there was a significant difference between the SUV_max_ in HER2-positive and HER2-negative lesions (Figure 4 and Figure 5). ROC analysis (Figure 6) showed that the use of the SUV_max_ threshold value of 5.36 for primary tumours would provide 100% sensitivity and 100% specificity. In the case of metastatic lymph nodes, 92% sensitivity and 100% specificity are expected when a cut-off SUV_max_ of 4.22 is applied. Thus, the measurement of SUV_max_ should be included in further evaluation plans as a parameter enabling the highest sensitivity and specificity of HER2 imaging. 

The results of this study show that the use of lesion-to-reference tissue ratios provides lower sensitivity and specificity compared to the measurements of SUVmax. However, the sensitivity and specificity of determining HER2 status in metastases using tumour-to-spleen ratios (83% and 100%, respectively; a cut-off value of 4.7) were better than the sensitivity and specificity using [^89^Zr]-trastuzumab-PET for extrahepatic lesions (per-patient basis) calculated using SUV_max_ (75.8% and 61.5%, respectively; cut-off value of 3.2) [35]. Taking this into account, it might make sense to include calculations of tumour-to-spleen ratios as a secondary outcome in a future Phase II study. This might provide a rationale for the use of [^99m^Tc]Tc-ADAPT6 in the case of stand-alone SPECT scanners in LMIC.

The high sensitivity and specificity in the discrimination of HER2-positive and HER2-negative lesions, which was demonstrated in this study, seem to be in contrast with the findings of some other clinical studies evaluating probes for radionuclide imaging of HER2 expression. For example, no clear correlation between SUV and immunohistochemistry data was observed in the evaluation of ^68^Ga-labelled HER2-specific single-domain antibodies [36]. A clinical evaluation of another HER2-specific single-domain antibody revealed an overlapping of SUV for HER2-positive and HER2-negative lesions [37]. In both cases, uptake of these tracers was quite high in tumours with low HER2 expression according to the biopsies. The authors claim that this is due to the heterogeneity of HER2 expression in tumours. This might be a correct explanation. However, we have to keep in mind that clinically HER2-negative breast cancer might express tens of thousands of receptors per cell [38]. Both cited studies utilized a low injected mass of sdAb (50–100 µg). Our experience with both ADAPT6 [22] and other HER2-targeting scaffold proteins, including affibody [26,30] and references therein and DARPins [33], demonstrated that injections of masses that were too low resulted in high uptake in tumours with low HER2 expression. Performing injected mass studies is essential to reach the best discrimination between tumours with high and low HER2 expression. 

Another possible problem might be caused by the expression of HER2 by normal hepatocytes [39]. This might result in the sequestering of an imaging probe in the liver when the injected mass is too low. This has been shown for the ^89^Zr-labelled anti-HER2 antibody trastuzumab [40]. A similar effect was also observed for radiolabeled antibodies targeting EGFR [41] and PSMA [42]. Apparently, this would complicate the visualization of liver metastases. Our experience [22] shows that an optimal injected mass of ADAPT6 leads to an increase in the tumour-to-liver ratio.

## 5. Conclusions

In conclusion, the use of SUV_max_ provided the best discrimination between HER2-positive and HER2-negative breast cancer lesions by SPECT/CT using [^99m^Tc]Tc-ADAPT6. Determination of the HER2 status of metastases using lesion-to-spleen ratios provides lower sensitivity but should be evaluated to indicate if it could be applied to imaging using stand-alone SPECT cameras that do not permit SUV calculations. 

## Figures and Tables

**Figure 1 pharmaceutics-16-00445-f001:**
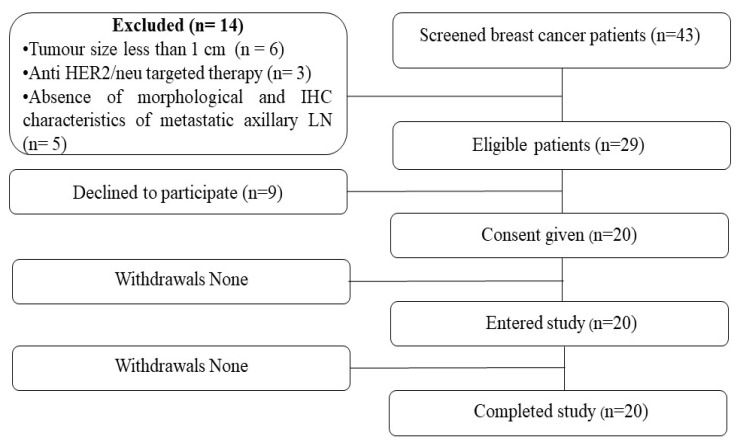
Flow diagram according to the Standards for Reporting of Diagnostic Accuracy Studies (STARD).

**Figure 2 pharmaceutics-16-00445-f002:**
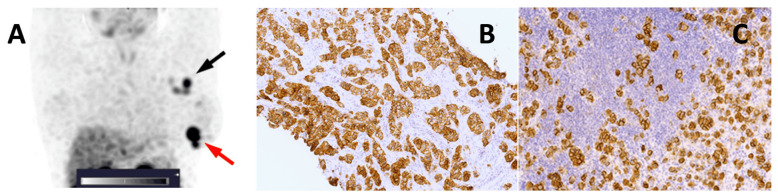
(**A**) A SPECT image of the primary tumour and a metastatic axillary lymph node in a HER2-positive breast cancer patient 2 h after injection of 500 µg of [^99m^Tc]Tc-ADAPT6; the red arrow points at the primary tumour; the black arrow points at the mALN. A linear SUV scale from 0 to 5 was used. (**B**) IHC staining of HER2 in the primary tumour (3+). (**C**) IHC staining of HER2 in the mALN.

**Figure 3 pharmaceutics-16-00445-f003:**
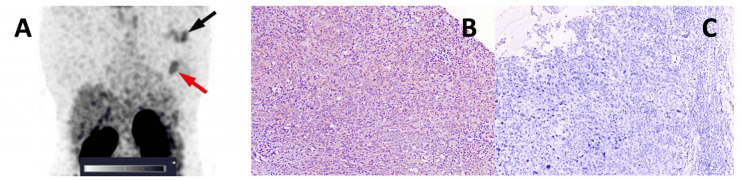
(**A**) A SPECT image of primary tumour and a metastatic axillary lymph node in a HER2-negative breast cancer patient 2 h after injection of 500 µg of [^99m^Tc]Tc-ADAPT6; the red arrow points at the primary tumour; the black arrow points at the mALN. A linear SUV scale from 0 to 5 was used. (**B**) IHC staining of HER2 in the primary tumour (0). (**C**) IHC staining of HER2 in the mALN.

**Figure 4 pharmaceutics-16-00445-f004:**
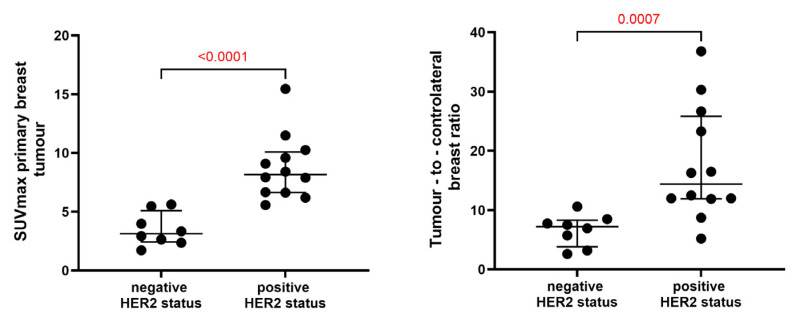
SUV_max_ in primary tumour and tumour-to-contralateral breast ratios in breast cancer patients 2 h after injection of [^99m^Tc]Tc-ADAPT6.

**Figure 5 pharmaceutics-16-00445-f005:**
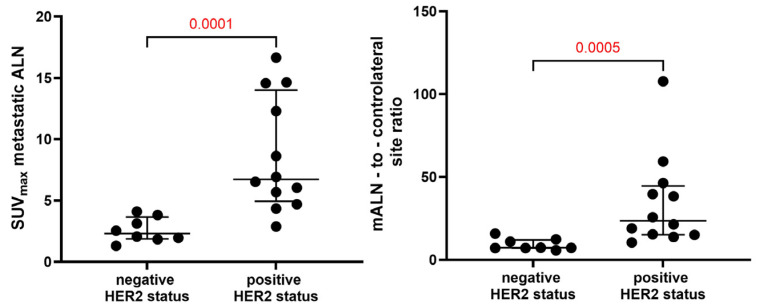
SUV_max_ in mALNs and mALN-to-contralateral site ratios in breast cancer patients 2 h after injection of [^99m^Tc]^99m^Tc-ADAPT6.

**Figure 6 pharmaceutics-16-00445-f006:**
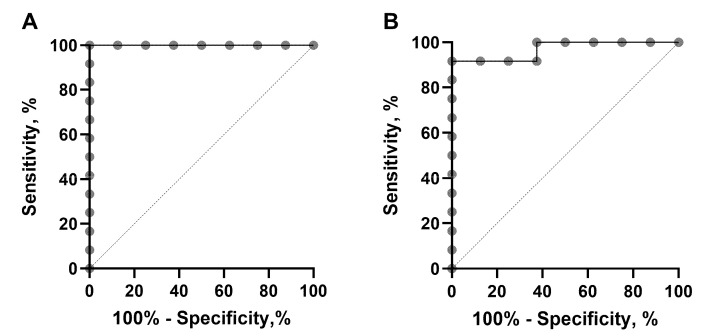
The receiver operating characteristic (ROC) curve analysis used to estimate the cut-off of [^99m^Tc]Tc-ADAPT6 uptake (SUV max) in primary breast cancer (**A**) and metastatic lymph nodes (**B**).

**Table 1 pharmaceutics-16-00445-t001:** Patient characteristics before injection with [^99m^Tc]Tc-ADAPT6.

No.	Age (y)	HER2 Status in Primary Tumour before Imaging (IHC *)	Primary Tumour Status (ER/PgR *)	HER2 Status in Axillary LN before Imaging (IHC)	Axillary LN Status (ER/PgR *)	Clinical Stage before Imaging	Tumor Size (mm)	mALN * (mm)
1 **	48	3+	ER+/PgR+	3+	ER+/PgR+	IIB (T2N1M0)	37	23
2	55	3+	ER−/PgR−	3+	ER+/PgR+	IIB (T2N1M0)	28	16
3 **	26	3+	ER+/PgR+	3+	ER+/PgR+	IIB (T2N1M0)	49	20
4	41	3+	ER+/PgR+	3+	ER+/PgR+	IIA (T2N1M0)	22	24
5	62	1+	ER+/PgR+	1+	ER+/PgR+	IIB (T2N1M0)	34	20
6 **	65	3+	ER+/PgR+	3+	ER+/PgR+	IIB (T2N1M0)	21	37
7	62	1+	ER+/PgR+	1+	ER+/PgR+	IIB (T2N1M0)	55	32
8 **	55	3+	ER−/PgR−	3+	ER−/PgR−	IIB (T2N1M0)	28	30
9	42	1+	ER+/PgR+	1+	ER+/PgR+	IIA (T2N1M0)	21	25
10	38	3+	ER+/PgR+	3+	ER+/PgR	IIB (T2N1M0)	38	12
11	47	1+	ER+/PgR+	1+	ER+/PgR+	IIB (T2N1M0)	38	13
12	47	1+	ER+/PgR+	1+	ER+/PgR+	IIB (T2N1M0)	23	32
13 **	61	3+	ER+/PgR+	3+	ER+/PgR+	IIB (T3N2M0)	30	32
14	59	3+	ER+/PgR+	3+	ER+/PgR+	IIB (T2N1M0)	29	16
15	38	3+	ER+/PgR+	3+	ER+/PgR+	IIB (T2N1M0)	21	26
16	54	1+	ER+/PgR−	1+	ER+/PgR−	IV (T3N2M1)	75	47
17	19	1+	ER+/PgR+	1+	ER+/PgR+	IIB (T2N1M0)	21	17
18	37	3+	ER−/PgR−	3+	ER−/PgR−	IIB (T2N1M0)	23	15
19	59	1+	ER+/PgR+	1+	ER+/PgR+	IIB (T2N1M0)	36	24
20	51	3+	ER+/PR+	3+	ER+/PgR+	IV (T4N3M1)	44	53

* IHC = immunohistochemistry; ER = estrogen receptor; PgR = progesterone receptor; mALNs = metastatic axillary lymph nodes; ** uptake values for these patients were also reported in [23].

## Data Availability

The data generated during the current study are available from the corresponding author upon reasonable request.

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
