# Peer review of "Evaluation of Approaches for the Assessment of HER2 Expression in Breast Cancer by Radionuclide Imaging Using the Scaffold Protein [99mTc]Tc-ADAPT6"

_pharmaceutics, 2024, doi:10.3390/pharmaceutics16040445_

Round 1

Reviewer 1 Report

Comments and Suggestions for Authors

The manuscript entitled "Evaluation of approaches for assessment of HER2-expression in breast cancer by radionuclide imaging using the scaffold protein [99mTc]Tc-ADAPT6" describes the optimization of the parameter(s) of [99mTc]Tc-ADAPT6 SPECT imaging to distinguish breast cancers with high and low expression of HER2. The study found that: the use of SUVmax provided the best discrimination between HER2 positive and HER2-negative breast cancer lesions by [99mTc]Tc-ADAPT6 based SPECT imaging; while lesion-to-spleen ratios provides lower sensitivity. The manuscript was well organized, and the literature citation was also appropriate. A few specific comments are listed below:

1. Since early-phase studies of [99mTc]Tc-ADAPT6 had been published by the same group (Cancers 2023, 15, 3149; J Nucl Med 2021; 62:493), please clarify the novelty and significance of this manuscript.

2. It seemed that some of patients in the current study (Table I) had been reported in their previous publications (e.g., Cancers 2023, 15, 3149; J Nucl Med 2021; 62:493). If so, please note those patients in the Table I.

3. The aim of this study is to “select the optimal parameter(s)”, and what are those selected optimal parameters? In addition, if only one optimal parameter was selected, please use “select the optimal parameter” in line 25; otherwise, please use “optimal parameters”.

Author Response

Reviewer 1

Open Review

(x) I would not like to sign my review report

( ) I would like to sign my review report

Quality of English Language

(x) I am not qualified to assess the quality of English in this paper

( ) English very difficult to understand/incomprehensible

( ) Extensive editing of English language required

( ) Moderate editing of English language required

( ) Minor editing of English language required

( ) English language fine. No issues detected

Yes         Can be improved              Must be improved           Not applicable

Does the introduction provide sufficient background and include all relevant references?

(x)           ( )           ( )           ( )

Are all the cited references relevant to the research?

(x)           ( )           ( )           ( )

Is the research design appropriate?

(x)           ( )           ( )           ( )

Are the methods adequately described?

( )           (x)           ( )           ( )

Are the results clearly presented?

( )           (x)           ( )           ( )

Are the conclusions supported by the results?

(x)           ( )           ( )           ( )

Comments and Suggestions for Authors

The manuscript entitled "Evaluation of approaches for assessment of HER2-expression in breast cancer by radionuclide imaging using the scaffold protein [99mTc]Tc-ADAPT6" describes the optimization of the parameter(s) of [99mTc]Tc-ADAPT6 SPECT imaging to distinguish breast cancers with high and low expression of HER2. The study found that: the use of SUVmax provided the best discrimination between HER2 positive and HER2-negative breast cancer lesions by [99mTc]Tc-ADAPT6 based SPECT imaging; while lesion-to-spleen ratios provides lower sensitivity. The manuscript was well organized, and the literature citation was also appropriate. A few specific comments are listed below:

  1. Since early-phase studies of [99mTc]Tc-ADAPT6 had been published by the same group (Cancers 2023, 15, 3149; J Nucl Med 2021; 62:493), please clarify the novelty and significance of this manuscript.

Answer. VT response proposal:

Phase I study (J Nucl Med 2021; 62:493 ) addressed questions of safety, tolerability and dosimetry. In addition, an optimal dose was found. Still, SUV values could not be determined in the phase I study because it has been performed using a stand-alone SPECT camera without CT.

The study published in Cancers 2023, 15, 3149 was dedicated to the comparison of two different scaffold proteins, 99mTc-ADAPT6 and 99mTc-DARPin G3 and addressed the uptake of both tracers in HER2-positive primary lesions to find the tracer providing highest tumour uptake and highest imaging contrast. As we pointed out in the Discussion “ Since an initial evaluation suggested that [99mTc]Tc-ADAPT6 provides higher uptake (SUVmax) in HER2-positive breast cancer lesions than 99mTc-labeled DARPin G3 [41], [99mTc]Tc-ADAPT6 was selected for further clinical evaluation.” Further, the second clinical study on 99mTc-ADAPT6 did not address the discrimination between HER2-positive and HER2-negative lesions.

This study is an extension of the second study addressing the sensitivity and specificity of the imaging. For this purpose, a cohort of HER2-negative patients has been included. Since the question concerned imaging of lymph node metastases, the patients without detectable metastases were excluded, but patients with HER2-positive lymph node metastases were included in this study.

This has now been further clarified in the introduction.

  1. It seemed that some of patients in the current study (Table I) had been reported in their previous publications (e.g., Cancers 2023, 15, 3149; J Nucl Med 2021; 62:493). If so, please note those patients in the Table I.

Answer. Thank you for pointing this out! We have marked such patients in the Table 1.

  1. The aim of this study is to “select the optimal parameter(s)”, and what are those selected optimal parameters? In addition, if only one optimal parameter was selected, please use “select the optimal parameter” in line 25; otherwise, please use “optimal parameters”.

Answer. Thank you for pointing this out! It was not clear if there would be only one optimal parameter or more when we planned this study. This was reflected in the aim of the study. Actually, while the SUVmax is the best parameter, the lesion-to-spleen ratio is still usable for stand-alone scanner. However, we followed the reviewer's advice and re-written the text as:

“This study aimed at the selection of the optimal parameters for distinguishing between breast cancers with high and low expression of HER2 using [99mTc]Tc-ADAPT6 in a planned phase II study.“.

Reviewer 2 Report

Comments and Suggestions for Authors

The manuscript entitled “Evaluation of approaches for assessment of HER2-expression in breast cancer by radionuclide imaging using the scaffold protein [99mTc]Tc-ADAPT6” describes the use of the small protein-based radioconjugate [99mTc]Tc-ADAPT6 to determine the HER2 expression level in the primary tumor and in metastatic axillary lymph nodes in breast cancer patients as part of a clinical trial. Additionally, the authors investigated different SPECT image analyses for the evaluation of the HER2 expression, addressing the applicability of quantitative SPECT to all users.

Minor and major issues unfortunately indicate a certain degree of lack of attention to detail. The following amendments and additions have to be addressed to be considered for publication:

Major:

A)The Discussion is long and disconnected and should be rearranged. The authors introduce different concepts (i.e., theranostics, scaffold proteins) but the connections between them, and with the main results of the manuscript (discussed in the remaining section, lines 352-409), are unclear. In the current state, the entire first section of Discussion (line 313-351) could be removed without affecting the narrative. If the authors wish to keep it, it should be reduced and linked in a clearer way.

B)   Material and Methods: The IHC procedure is missing. The authors should add the description of the method or a suitable reference with the detailed procedure.

C)   Material and Methods: Was the radioactive agent prepared in a GMP compliant way? There is no description of the preparation of the agent in the manuscript. The reference given by the authors did not describe the procedure either but sent the reader to another reference paper with the description of the preparation of the product for pre-clinical studies. If existing, the authors should indicate a more suitable reference for the cGMP procedure. If not provided in previous publications, the authors should complement, in this manuscript, the referenced pre-clinical preparation with the main details of the cGMP preparation including the list of QC methods and acceptance specifications of the radioactive product for human administration. Just a paragraph in the Material and Methods would be sufficient.

D)   Figure 2 and 3: The figures have multiple issues. Initially, they show H&E and not IHC staining. The authors should provide the right IHC images. Secondly, the SPECT images used for the HER2+ and HER2- patients are the same, as well as the two provided H&E figures (although one, for some reason, is more blurred than the other).

E)   The authors mention multiple time a Fig.6 (page 8, line 285; Page 9, line 304; page 11, line 365) which is missing from the manuscript. Please add it.

Minor:

The English throughout the whole manuscript is good but there are multiple typos, imprecisions (e.g., random parentheses, missing letters), repetitions, and punctuation errors.

Page 3, line 143: the abbreviation mALN is explained later on (page 5, table 1) but should explained at this point.

Table 1 has multiple typos, spurious footer symbol/letter, and missing terms and letters.

Page 11, lines 410-414: The conclusions (word by word) are repeated twice.

Page 3 lines 137-140 and Page12 lines 447-450: the clinical trial approval and identifier are repeated in two places. The Material and Methods might be the most suitable (because more visible) place for such details.

Supplemental figures and tables numbering in the main text should include the S (for consistency with the Supplemental Information); e.g., Supplemental Table S2. Additionally, there is no Supplemental Fig.6 (page 9, line 311).

Page 10, line 335: Repetition of “in LMIC countries”.

Page 10, line 338: The general sentence “agent, such as small proteins” might be more suitable than the very specific, and therefore limiting, term “scaffold protein”.

References: There are many typos, punctuation errors, inconsistencies (titles with or without capital letters).

The journal name, year, and DOI of reference 35 are wrong! Also, the second author’s surname of reference 35 is “de Vries”, not just “Vries”. The authors should thoroughly check that the references are correct.

Comments on the Quality of English Language

The English throughout the whole manuscript is good. There are just typos, imprecisions (e.g., random parentheses, missing letters), repetitions, and punctuation errors

Author Response

Reviewer 2

Open Review

(x) I would not like to sign my review report

( ) I would like to sign my review report

Quality of English Language

( ) I am not qualified to assess the quality of English in this paper

( ) English very difficult to understand/incomprehensible

( ) Extensive editing of English language required

( ) Moderate editing of English language required

(x) Minor editing of English language required

( ) English language fine. No issues detected

Yes         Can be improved              Must be improved           Not applicable

Does the introduction provide sufficient background and include all relevant references?

(x)           ( )           ( )           ( )

Are all the cited references relevant to the research?

(x)           ( )           ( )           ( )

Is the research design appropriate?

(x)           ( )           ( )           ( )

Are the methods adequately described?

( )           ( )           (x) Must be improved      ( )

Are the results clearly presented?

( )           (x) Are the results clearly presented?       ( )           ( )

Are the conclusions supported by the results?

(x)           ( )           ( )           ( )

Comments and Suggestions for Authors

The manuscript entitled “Evaluation of approaches for assessment of HER2-expression in breast cancer by radionuclide imaging using the scaffold protein [99mTc]Tc-ADAPT6” describes the use of the small protein-based radioconjugate [99mTc]Tc-ADAPT6 to determine the HER2 expression level in the primary tumor and in metastatic axillary lymph nodes in breast cancer patients as part of a clinical trial. Additionally, the authors investigated different SPECT image analyses for the evaluation of the HER2 expression, addressing the applicability of quantitative SPECT to all users.

Minor and major issues unfortunately indicate a certain degree of lack of attention to detail. The following amendments and additions have to be addressed to be considered for publication:

Major:

  1. A) The Discussion is long and disconnected and should be rearranged. The authors introduce different concepts (i.e., theranostics, scaffold proteins) but the connections between them, and with the main results of the manuscript (discussed in the remaining section, lines 352-409), are unclear. In the current state, the entire first section of Discussion (line 313-351) could be removed without affecting the narrative. If the authors wish to keep it, it should be reduced and linked in a clearer way.

Answer. We apologise for the unclear text. We tried to present our reasoning but (apparently) failed. Therefore, this first part of the Discussion section has been deleted.

  1. B) Material and Methods: The IHC procedure is missing. The authors should add the description of the method or a suitable reference with the detailed procedure.

Answer. We apologise for the unclear text in the original submission. Unfortunately, a plagiarism checking system marks nearly all descriptions in Materials and Methods as (self)plagiarism. Therefore, we tried to keep this as short as possible. Apparently, this makes the text much less clear. In the revised version, we have made a subsection in the Materials and Methods “Immunohistochemistry analysis” describing this in more detail.

“In all patients the core biopsies of the primary tumours and the mALN were performed under ultrasound guidance and the HER2 expression was evaluated by immunohistochemistry (IHC). Formalin-fixed paraffin-embedded sections (7 µm) were stained by VENTANA anti-HER-2/neu (4B5) Rabbit Monoclonal Primary Antibody using Ventana Benchmark Ultra Instrument (Roche) according to the manufacturer's instructions. HER2 expression was scored according to the guidelines of the American Society of Clinical Oncology and the College of American Pathologists (ASCO/CAP2018) [8]. A score of 3+ by IHC was defined as HER2-positive status. In cases of equivocal IHC status, FISH test (HER2/CEP17 FISH probes, Kreathech) was performed according to the manufacturer's instructions. HER2 status was considered as negative in case of a score of 0 and 1+ by IHC or score 2+ and FISH negative.“.

  1. C) Material and Methods: Was the radioactive agent prepared in a GMP compliant way? There is no description of the preparation of the agent in the manuscript. The reference given by the authors did not describe the procedure either but sent the reader to another reference paper with the description of the preparation of the product for pre-clinical studies. If existing, the authors should indicate a more suitable reference for the cGMP procedure. If not provided in previous publications, the authors should complement, in this manuscript, the referenced pre-clinical preparation with the main details of the cGMP preparation including the list of QC methods and acceptance specifications of the radioactive product for human administration. Just a paragraph in the Material and Methods would be sufficient.

Answer. To address this comment, we have added the following text to the manuscript

“Purification and labelling of ADAPT6

ADAPT6 protein was produced in E. coli and purified using immobilized metal ion chromatography as described in [20]. Analysis by liquid chromatography-electrospray ionization mass spectrometry (6520 Accurate Q-TOF LC/ MS, Agilent) confirmed the identity of the protein (measured Mw 6954 Da, calculated Mw 6954.7 Da). No impurities were detected by reversed-phase HPLC (RP-HPLC) using  Zorbax 300SB-C18 column (4.6 × 150 mm, 3.5 μm particle size, Agilent), i.e. chemical purity was 100%. The levels of endotoxins (0.49 EU/mg freeze-dried protein) and residual E coli proteins (31.8 ng/mg of freeze-dried protein) were very low and met the requirements of European Pharmacopeia. Aliquots containing 500 µg ADAPT6 were prepared and freeze-dried.

Test labelling of qualification batches was performed according to the protocol described below. The identity of ADAPT6 labelled with 99mTc was confirmed by radio-RP-HPLC (Phenomenex LC Luna 5 µm C18 column, 150 x 4.6 mm, 100 Å particle size, Danaher). Specific binding of the clinical batch of [99mTc]Tc-ADAPT6 to HER2-expressing cancer cells was confirmed by in vitro saturation assay, as described in [20]. Sterility and endotoxin levels were evaluated according to the European Pharmacopoeia after decay of 99mTc. According to national guidelines for conducting preclinical studies of drugs, the single-dose toxicity after intravenous injection was determined in mice and rats. No toxic effects were observed.

Radiolabelling was performed in a GMP-compliant way at the Department of Radionuclide Therapy and Diagnostics, Tomsk Сancer Research Institute, according to national regulations. Freeze-dried ADAPT6 (500 µg) was reconstituted by adding sterile sodium phosphate buffer, pH 7.5 (100 µL) using a sterile syringe followed by incubation for 30 min at room temperature. An eluate from a generator of 99mTc (500 µL) is added to a sealed vial containing the CRS kit (Centrum for Radiopharmaceutical Sciences, Willigen, Switzerland) and incubated at 100°C for 30 minutes. After incubation, 400 μL of the resulting solution was transferred by a sterile syringe to the vial containing reconstituted ADAPT6, followed by incubation for 60 min at 50°C.  [99mTc]Tc-ADAPT6 was purified by size exclusion chromatography using sterilized NAP-5 columns (Sephadex G-25, GE, Healthcare, USA) pre-equilibrated and eluted with sterile sodium phosphate buffer. The purified fraction is brought to a volume of 10 mL using sterile isotonic NaCl solution.

A small aliquot was taken for analysis of pH and radiochemical purity. The pH of the drug product was determined by pH test strips.  Routine analysis of the radiochemical purity was performed using instant thin layer chromatography (Agilent Technologies, Santa Clara, CA, USA). The mobile phases were PBS (Rf = 0 for [99mTc]Tc-ADAPT6 and [99mTc]TcO2; Rf = 1 for [99mTc]Tc(H2O)3(CO)3+ and [99mTc]TcO) and pyridine:acetic acid:water, 10:6:3 (Rf = 0 for [99mTc]TcO2 and Rf = 1 for the [99mTc]Tc-ADAPT6, [99mTc]Tc(H2O)3(CO)3 and [99mTc]TcO4-). The bubble-point method was used to test the filter integrity.

A visual inspection was performed. The solution was clear, non-opalescent and colourless. The radiochemical purity of [99mTc]Tc-ADAPT6 was 97.6 ± 1.4 %. The pH was 7.4 Acceptance criteria of radiochemical purity, activity concentration, activity, pH, colour/transparency, and endotoxin level were met.

  1. D) Figure 2 and 3: The figures have multiple issues. Initially, they show H&E and not IHC staining. The authors should provide the right IHC images. Secondly, the SPECT images used for the HER2+ and HER2- patients are the same, as well as the two provided H&E figures (although one, for some reason, is more blurred than the other).

Answer. We apologise for this late night work artefact during preparation of the file for a submission. The figure shows only HER2-negative lesions, thus only H&E counterstaining is visible.  A correct figure has been pasted in the revised version.

  1. E) The authors mention multiple time a Fig.6 (page 8, line 285; Page 9, line 304; page 11, line 365) which is missing from the manuscript. Please add it.

Answer. Thank you for pointing this out! Figure 6 has been added to the revised version of the manuscript.

Minor:

The English throughout the whole manuscript is good but there are multiple typos, imprecisions (e.g., random parentheses, missing letters), repetitions, and punctuation errors.

Answer. Thank you! The text was re-edited.

Page 3, line 143: the abbreviation mALN is explained later on (page 5, table 1) but should explained at this point.

Answer. Thank you for pointing this out! This has been corrected in the revised version according to the reviewer’s suggestion.

Table 1 has multiple typos, spurious footer symbol/letter, and missing terms and letters.

Answer. Thank you for pointing at this! This has been corrected in the revised version.

Page 11, lines 410-414: The conclusions (word by word) are repeated twice.

Answer. Thank you for pointing at this! The conclusions have been removed from the Discussion and remain only in the Conclusions section.

Page 3 lines 137-140 and Page12 lines 447-450: the clinical trial approval and identifier are repeated in two places. The Material and Methods might be the most suitable (because more visible) place for such details.

Answer. Thank you for raising this question. We absolutely agree that the Material and Methods might be the most suitable place for such details. However, the Institutional Review Board Statement and approval number for studies involving humans must be placed in the Back Matter, under “Institutional Review Board Statement”, according to the Instructions for Authors. Therefore, we have removed the clinical trial identifier from the Back Matter, but decided to keep Institutional Review Board Statement both in the Material and Methods and in the Material and Methods.

Supplemental figures and tables numbering in the main text should include the S (for consistency with the Supplemental Information); e.g., Supplemental Table S2. Additionally, there is no Supplemental Fig.6 (page 9, line 311).

Answer. Thank you for pointing at this! This has been corrected in the revised version. Concerning “Supplemental Fig. 6”, We have corrected this to “(Supplementary Fig. S4).” in the revised version.

Page 10, line 335: Repetition of “in LMIC countries”.

Answer. Thank you for pointing at this! This has been corrected in the revised version.

Page 10, line 338: The general sentence “agent, such as small proteins” might be more suitable than the very specific, and therefore limiting, term “scaffold protein”.

Answer. Thank you for pointing at this! The text in the revised version has been modified according to reviewer’s suggestion:

“A short time interval between the injection of small targeting proteins and imaging [38] makes it possible to use relatively cheap and readily available 99mTc for labelling.“.

References: There are many typos, punctuation errors, inconsistencies (titles with or without capital letters).

Answer. Thank you for pointing at this! This has been corrected in the revised version.

The journal name, year, and DOI of reference 35 are wrong! Also, the second author’s surname of reference 35 is “de Vries”, not just “Vries”. The authors should thoroughly check that the references are correct.

Answer. Thank you for pointing at this! This has been corrected in the revised version.

Round 2

Reviewer 2 Report

Comments and Suggestions for Authors

The authors has answered to the queries and have implemented the corrections/additions in a satisfactory way